# Systematic review of clinical outcome reporting in randomised controlled trials of burn care

Amber E Young,[1] Anna Davies,[1] Sophie Bland,[2] Sara Brookes,[1,3] Jane M Blazeby[1]

¹Population Health Sciences, Bristol Medical School, University of Bristol, Bristol, UK
²Patient representative, Bristol, Avon, UK
³Cancer Research UK Clinical Trials Unit (CRCTU), Institute of Cancer and Genomic Sciences, University of Birmingham, Birmingham, UK

**Correspondence to**
Dr Amber E Young;
amber.young1@nhs.net

## ABSTRACT

**Introduction** Systematic reviews collate trial data to provide evidence to support clinical decision-making. For effective synthesis, there must be consistency in outcome reporting. There is no agreed set of outcomes for reporting the effect of burn care interventions. Issues with outcome reporting have been identified, although not systematically investigated. This study gathers empirical evidence on any variation in outcome reporting and assesses the need for a core outcome set for burn care research.

**Methods** Electronic searches of four search engines were undertaken from January 2012 to December 2016 for randomised controlled trials (RCTs), using medical subject headings and free text terms including 'burn', 'scald' 'thermal injury' and 'RCT'. Two authors independently screened papers, extracted outcomes verbatim and recorded the timing of outcome measurement. Duplicate outcomes (exact wording ± different spelling), similar outcomes (albumin in blood, serum albumin) and identical outcomes measured at different times were removed. Variation in outcome reporting was determined by assessing the number of unique outcomes reported across all included trials. Outcomes were classified into domains. Bias was reduced using five researchers and a patient working independently and together.

**Results** 147 trials were included, of which 127 (86.4%) were RCTs, 13 (8.8%) pilot studies and 7 (4.8%) RCT protocols. 1494 verbatim clinical outcomes were reported; 955 were unique. 76.8% of outcomes were measured within 6 months of injury. Commonly reported outcomes were defined differently. Numbers of unique outcomes per trial varied from one to 37 (median 9; IQR 5,13). No single outcome was reported across all studies demonstrating inconsistency of reporting. Outcomes were classified into 54 domains. Numbers of outcomes per domain ranged from 1 to 166 (median 11; IQR 3,24).

**Conclusions** This review has demonstrated heterogeneity in outcome reporting in burn care research which will hinder amalgamation of study data. We recommend the development of a Core Outcome Set.

**PROSPERO registration number** CRD42017060908.

## Strengths and limitations of this study

► This review is a comprehensive and systematic search for all clinical outcomes reported in randomised controlled trials of burn care between and including 2012 and 2016.
► There is a detailed analysis of all reported outcomes and timing of outcome assessment.
► A multidisciplinary team including a patient were involved in the study.
► Quality assessment of studies was not undertaken as the purpose of the review was to extract clinical outcomes alone and not to assess the effect of an intervention.

## INTRODUCTION

Each year an estimated 2–300 000 people die from burn injuries globally.[1] Millions more suffer from burn-related disabilities and disfigurements.[2] These injuries have functional, psychological, social and economic effects on survivors and their families. There are multiple strategies for managing burn wounds and the associated impact on patient physiology, with new care pathways and technology being introduced on a regular basis.[3–5] The choice of treatment should be made using up-to-date, high quality scientific evidence.[6 7] Systematic reviews of randomised controlled trials (RCT) are regarded as the highest quality evidence.[8–10] Despite increasing numbers of published RCTs in burn care, systematic reviews have not provided evidence to support many commonly used interventions or management strategies.[11–13]

A well-designed RCT requires that outcomes are prespecified. Evidence synthesis requires that these outcomes are consistent across RCTs in the same healthcare area.[14] In the context of clinical trials, Williamson *et al* in the Core Outcome Measures in Effectiveness Trials (COMET) handbook, define an outcome as 'a measurement or observation used to capture and assess the effect of treatment such as assessment of side effects (risk) or effectiveness (benefits)'.[15] Chan adds a temporal element: 'a variable measured at a specific time point to assess the efficacy or harm of an intervention'.[16] If RCTs report outcomes that cannot be collated due to differences in choice, definition or timepoint

of assessment, evidence synthesis will not be effective or efficient. There is no agreed minimum set of outcomes important to patients and professionals for reporting in burn care trials and problems with outcome reporting in burn care research have previously been suggested.[17–19]

Prespecifying outcomes requires research to determine and agree the most important outcomes for a clinical condition. If this is not undertaken, the outcomes reported may not reflect patients' or other stakeholders' needs, outcomes will vary between studies (outcome reporting heterogeneity), and it will be difficult to determine if authors have reported all the outcomes they measured (outcome reporting bias).[20 21] Choosing the most important outcomes to measure in burn care is complex, as patients are a heterogeneous population, with variations in age, mechanism of injury, depth, site and size of burn.[22 23] The time frame at which outcomes are measured may also determine the types of outcomes assessed. Outcomes reported in clinical trials during the acute treatment phase include healing time, skin-graft loss, infection rates and National Health Service costs.[24–27] Longer-term reported outcomes relate to functional, cosmetic and psychological issues.[28]

To date, there has been no formal investigation into outcome reporting in trials of burn care. The purpose of this study is to examine clinical outcome reporting in burn care research, to consider the types, definitions and timing of outcomes measured and to consider the need for a Core Outcome Set (COS) in this field.

## METHODS

This review is focused on clinical, observer-reported outcomes in RCTs assessing the impact of interventions in burn care. It adhered to a prespecified protocol and the Preferred Reporting Items for Systematic Reviews and Meta-Analyses (PRISMA) statement.[29]

### Study eligibility

Studies were included if they met the following:

*Types of studies*: We included full text RCTs along with RCT protocols and pilot studies. The study design was limited to RCTs, as any final COS will be used for RCT reporting.[29] We excluded protocols and pilot studies if the full RCT had been published within the selected time period. We also excluded conference proceedings and abstracts, non-English language publications and studies not involving human subjects.

*Types of participants*: We included studies recording outcomes from patients of any age with a cutaneous burn of any type or size, determined by either clinician evaluation or objective assessment, or both, which required treatment in any healthcare facility. Studies where the population consisted of patients with combined thermal and mechanical injuries were only included if it was possible to separate out the burn care outcomes. Trials studying patients with pure carbon monoxide poisoning, chemical ocular or caustic oesophageal burns were

excluded, as the former does not involve a burn and the latter have different aetiology and management to cutaneous burns.

*Type of interventions*: Any surgical or non-surgical burn care intervention with any appropriate *comparator*.

*Types of outcomes*: Defined as the exact terms used in a published trial abstract, methods or results including tables and figures for any observer-reported clinical endpoint. These included physiological, metabolic or adverse or mortality events measured by researchers and relevant to patients' recovery and long-term well-being after burn care.[30] Trials assessing quality of life were only included if the data were observer-reported.

### Identification of studies

Electronic searches of Ovid MEDLINE, Ovid EMBASE, Web of Science and The Cochrane Library were searched from 1 January 2012 to 31 December 2016 for RCTs related to burn care using medical subject heading and free text terms including 'burn', 'scald' 'thermal injury' and 'RCT'. This period was chosen so that the outcomes extracted, reflected use in trials relating to modern burn care. Limiting the review to 5 years allowed us to balance workload against the likelihood of selecting enough trials fulfilling inclusion criteria to demonstrate whether heterogeneity of outcome reporting was present in burn care research. The thesaurus vocabulary of each database was used to adapt the search terms. The search strategy for Ovid MEDLINE is included in a previous publication and in online supplementary appendix A.[29]

### Study selection process

Prior to both abstract and full-text screening, all review authors underwent training to ensure a comparable understanding of the purpose of the review and the eligibility criteria. The reference management software EndNote V.6 (Clarivate Analytics, Boston; available at http://endnote.com/) was used to compile all titles derived from the initial searches, with duplicates removed for the review authors to screen titles and abstracts against the eligibility criteria. Screening of titles and abstracts was completed independently, then in duplicate by two authors (AY, AD) with experience in systematic review methodology. All screening disagreements were discussed, with any outstanding disagreements resolved by an independent reviewer (JB). Any studies appearing to meet the inclusion criteria based on the abstract were retrieved as full-text articles. Two reviewers then read the full-text articles in their entirety to assess for eligibility, with decisions on inclusion and exclusion recorded (figure 1 for flow diagram). Reasons for exclusion were ordered hierarchically (box 1) and applied to each full text. The highest reason for exclusion met by a paper was recorded as its reason for exclusion. Any disagreements were discussed with another author (JB).

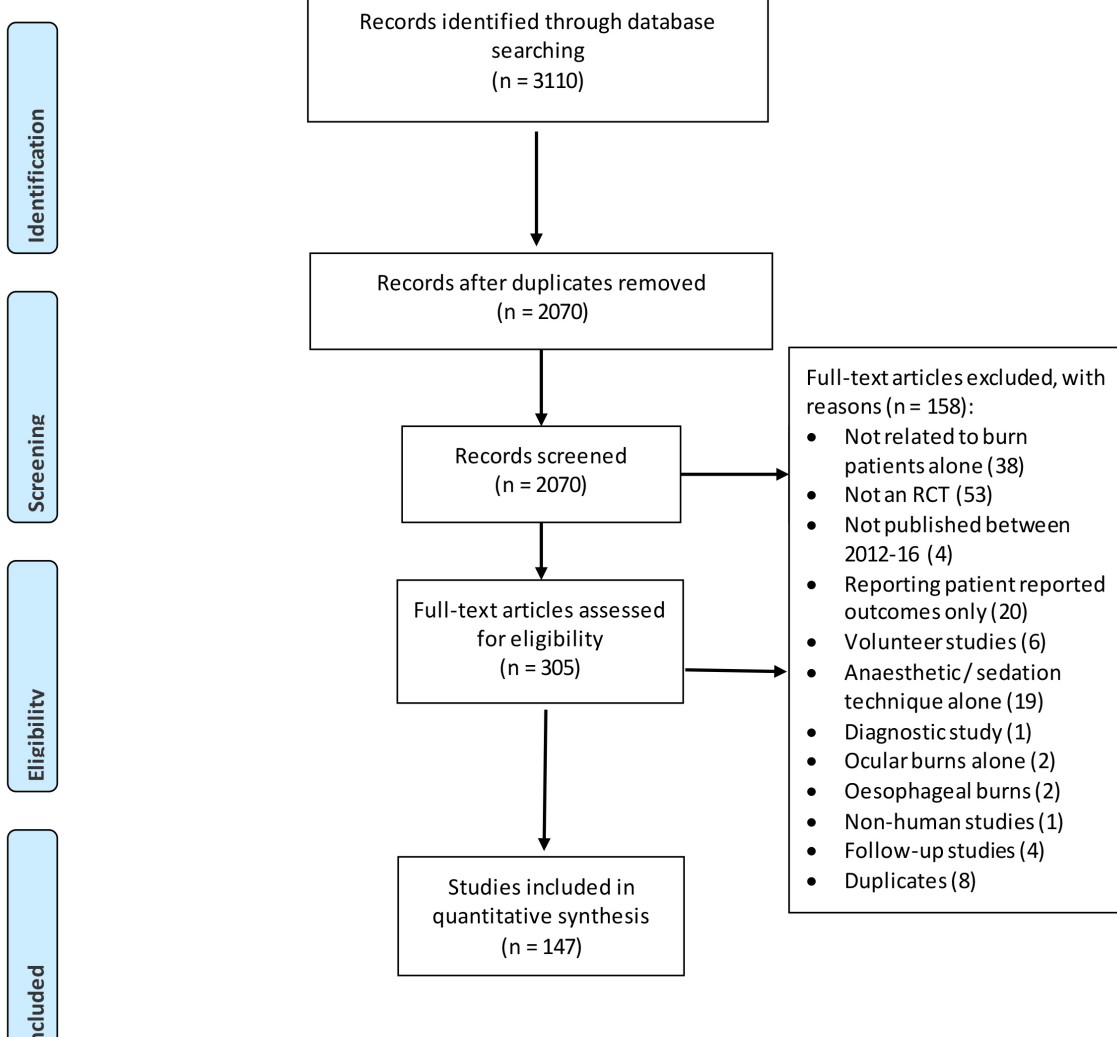

**Figure 1** Preferred Reporting Items for Systematic Reviews and Meta-Analyses flow diagram. PRO, patient reported outcome; RCT, randomised controlled trial.

## Quality assessment

The aim of this study was to comprehensively document any variation in clinical outcomes selected, defined, measured and reported in burn care RCTs and not to synthesise data about the effect of interventions. Inclusion of all trials was necessary to demonstrate if a variation in outcome reporting was present across trials, regardless of quality of methodology of the trial. We therefore decided not to undertake a quality assessment of studies because it was not relevant to the data being recorded in this review; simply the nature and description of the unique outcomes reported in each study.

## Data extraction

Data were extracted into a standardised data extraction sheet (Microsoft Excel). This included study author, country or countries recruiting (categorised into the United Nations six regions[31]), publication year, number of sites and number of participants recruited per trial, design (full RCT, pilot, protocol) and intervention tested.

For protocols, the planned participant inclusion criteria and sample size were extracted.

No distinction was made between primary or secondary outcomes, although this was noted and is part of a separate project. All outcomes were extracted verbatim, with 20% of the extracted data verified by a second reviewer. True duplicates, spelled and worded the same, were deleted. As a second process, two reviewers (a clinician and researcher) discussed all verbatim outcomes to assess duplicates in meaning but spelled or worded in a slightly different manner; such as length of time in hospital and number of days in hospital, platelet level and levels, and serum interleukin (IL) 10 and IL-10 in blood. These were named as one outcome with wording chosen by the reviewers and the others deleted as duplicates. The remaining outcomes were therefore all different in meaning. Any discrepancies were discussed with a senior researcher (JB). The number of outcomes per trial and the variation of outcomes between trials was recorded.

## Box 1 Hierarchy of exclusion

1. Duplicate.
2. Not published between 1 January 2012 and 31 December 2016.
3. Not written in English.
4. Not relating to burns or burn care alone.
5. Population is non-human subjects (eg, animal studies, modelling).
6. Abstract with no full text given.
7. Study is not a randomised controlled trial testing an intervention on burn care.
8. Laboratory based, not carried out in clinical setting, for example, exudate or blood samples from humans tested within laboratory.
9. Systematic review paper, commentary paper.
10. Reports only patient-reported outcomes.
11. Volunteer study.
12. Oesophageal burns.
13. Ocular burns.
14. Anaesthetic/sedation technique.
15. Diagnostic test trial.
16. Unavailable anywhere.
17. Follow-up studies.

The time after injury that outcomes were measured were noted separately in order to (a) assess the heterogeneity in outcome measurement timing and (b) to understand at what stage after injury the effects of the intervention were being assessed. If a single outcome was assessed at different timepoints, *all* assessment timings were recorded. Data extraction for the timing of outcome reporting from 10% of trials was undertaken independently by another researcher. Timings of outcome assessment were categorised into time periods; <1 month, >1 month and ≤3 months, >3 months and ≤6 months after injury, >6 months and ≤1 year and >1 year and ≤3 years, and >3 years. We reported two other outcome time periods; those assessed during acute hospitalisation and during burn wound healing, as these were commonly reported in the literature with no proscribed timepoint. However, it was clear from the reported length of stay and healing data, that all these outcomes were assessed within 6 months of injury. The frequency of outcomes reported within each time period was recorded.

The data were tabulated so that each study was listed with study and population details along with outcomes measured. Outcomes were extracted from this spreadsheet into another, with duplicates removed as described above. Outcomes measuring the same healthcare issue but at different timepoints were noted as one outcome for the final set. These final unique outcomes were then grouped into domains.

### Classification of outcomes into domains

Outcome domains are groups of similar outcomes. This organisation is necessary, as maintaining a large set of outcomes when a significant number are similar, would make any future classification of the outcomes in terms of importance, extremely challenging.

Outcomes were classified into domains in a three-stage iterative approach. In stage 1, four researchers (a clinically trained burn care researcher, a burn research associate and two senior research nurses experienced in burn care) independently reviewed the list of outcomes and attributed a potential domain to each one using their own terms. In stage 2, the researchers met to review the domains and agreed (1) appropriate groupings of outcomes into domains and (2) an appropriate name for each domain. Rules for attribution of outcomes to domains were recorded in a coding log to ensure consistency. In stage 3, a patient representative reviewed the outcomes and their attributed domains to check for clarity of domain name, and that the outcomes under each domain were appropriately attributed. A final meeting with an experienced outcome researcher was held to finalise outcomes and domains. The use of a published classification system was not undertaken as none appeared to allow the flexibility or fit to the types of outcomes reported in burn care trials.[32 33]

The results described below indicate the characteristics of the reported studies and provide detail on heterogeneity of outcome reporting between studies, outcome definitions, timepoints and outcome domains.

### Patient involvement

The need for a burn care COS project was conceived following discussions regarding clinical healthcare Key Performance Indicators with professionals and patients. The patients were vocal about outcomes important to them which they felt were overlooked by professionals, such as pain. The systematic review was discussed at regular project steering group meetings attended by three patients with burns and one parent of a child with burns. A patient with burns is a coauthor and was involved with writing and editing of this article as well as with the naming of the outcome domains. Dissemination will be to the lay representatives of the steering group and will inform the Core Outcome Set study in which patients are actively involved.

## RESULTS

### Included studies and study protocols

The initial search strategy identified 3110 studies. Following de-duplication, a total of 2070 studies remained. Independent scrutiny of the titles and abstracts identified 306 potentially relevant articles for full text review. Of these, 158 studies did not meet our inclusion criteria and were excluded (PRISMA flow diagram; figure 1). Therefore, a total of 147 studies formed the basis of this study.[24 34–178]

### Studies

Of the 147 studies (table 1), 86.4% (127) were reports of full RCTs, 8.8% (13) were pilot studies and 4.8% (7) were study protocols. The number of studies published increased between 2012 and 2016, with 26 RCTs published in 2012 and 40 in 2016 (table 2).

**Table 1** Included randomised controlled trials

| Trial title | First author | Year of publication |
|---|---|---|
| 1. Comparison of silver nylon wound dressing and silver sulfadiazine in partial burn wound therapy.[34] | Abedini | 2012 |
| 2. Healing of burn wounds by topical treatment: A randomized controlled comparison between silver sulfadiazine and nano-crystalline silver.[35] | Adhya | 2015 |
| 3. An analysis of deep vein thrombosis in burn patients (Part 1): Comparison of D-dimer and Doppler ultrasound as screening tools.[36] | Ahuja | 2016 |
| 4. An analysis of deep vein thrombosis in burn patients (part II): A randomized and controlled study of thrombo-prophylaxis with low molecular weight heparin.[37] | Ahuja | 2016 |
| 5. A four arm, double blind, randomized and placebo-controlled study of pregabalin in the management of post-burn pruritus.[38] | Ahuja | 2012 |
| 6. Propranolol attenuates haemorrhage and accelerates wound healing in severely burned adults.[39] | Ali | 2015 |
| 7. Aerobic exercise training in modulation of aerobic physical fitness and balance of burned patients.[40] | Ali | 2015 |
| 8. Silk sericin ameliorates wound healing and its clinical efficacy in burn wounds.[41] | Aramwit | 2013 |
| 9. A Randomized Controlled Trial Comparing Endoscopic-Assisted Versus Open Neck Tissue Expander Placement in Reconstruction of Post-Burn Facial Scar Deformities.[42] | As'adi | 2016 |
| 10. A prospective, randomised study of a novel transforming methacrylate dressing compared with a silver-containing sodium carboxymethylcellulose dressing on partial-thickness skin graft donor sites in burn patients.[43] | Assadian | 2015 |
| 11. Multimodal quantitative analysis of early pulsed-dye laser treatment of scars at a pediatric burn hospital.[44] | Bailey | 2012 |
| 12. Early fluid resuscitation with hydroxyethyl starch 130/0.4 (6%) in severe burn injury: a randomized, controlled, double-blind clinical trial.[45] | Bechir | 2013 |
| 13. A prospective randomized trial comparing silver sulfadiazine cream with a water-soluble poly-antimicrobial gel in partial-thickness burn wounds.[46] | Black | 2015 |
| 14. Clinical effectiveness of dermal substitution in burns by topical negative pressure: a multicenter randomized controlled trial.[47] | Bloeman | 2012 |
| 15. Effect of subcutaneous epinephrine/saline/local anesthetic versus saline-only injection on split-thickness skin graft donor site perfusion, healing, and pain.[48] | Blome Eberwein | 2013 |
| 16. A randomized controlled study of silver-based burns dressing in a pediatric emergency department.[49] | Brown | 2016 |
| 17. Cost-Effectiveness of a Nonpharmacological Intervention in Pediatric Burn Care.[50] | Brown | 2015 |
| 18. Play and heal: randomized controlled trial of DittoTM intervention efficacy on improving re-epithelialization in pediatric burns.[51] | Brown | 2013 |
| 19. The implementation and evaluation of therapeutic touch in burn patients: an instructive experience of conducting a scientific study within a non-academic nursing setting. | Busch | 2012 |
| 20. Prophylactic sequential bronchoscopy after inhalation injury: results from a 3 year prospective randomized trial.[52] | Carr | 2013 |
| 21. Burns injury in children: is antibiotic prophylaxis recommended?[53] | Chahed | 2014 |
| 22. A randomized controlled trial to compare the effects of liquid versus powdered recombinant human growth hormone in treating patients with severe burns.[54] | Chen | 2016 |
| 23. The Effect of Continuous Sedation Therapy on Immunomodulation, Plasma Levels of Antioxidants, and Indicators of Tissue Repair in Post-Burn Sepsis Patients.[55] | Chen | 2015 |
| 24. Application of acellular dermal xenografts in full-thickness skin burns.[56] | Chen | 2013 |
| 25. Effectiveness of medical hypnosis for pain reduction and faster wound healing in pediatric acute burn injury: study protocol for a randomized controlled trial.[57] | Chester | 2016 |
| 26. Safety of recombinant human granulocyte-macrophage colony-stimulating factor in healing paediatric severe burns.[58] | Chi | 2015 |
| 27. Comparison of three cooling methods for burn patients: A randomized clinical trial.[59] | Cho | 2016 |

Continued

**Table 1** Continued

| Trial title | First author | Year of publication |
|---|---|---|
| 28. The effect of burn rehabilitation massage therapy on hypertrophic scar after burn: a randomized controlled trial.[60] | Cho | 2014 |
| 29. Effect of extracorporeal shock wave therapy on scar pain in burn patients: A prospective, randomized, single-blind, placebo-controlled study.[61] | Cho | 2016 |
| 30. Characterization of early thermal burns and the effects of hyperbaric oxygen treatment: a pilot study.[62] | Chong | 2013 |
| 31. Effects of different duration exercise programs in children with severe burns.[63] | Clayton | 2016 |
| 32. The effect of healing touch on sleep patterns of pediatric burn patients.[64] | Cone | 2014 |
| 33. Effect of N-acetylcysteine treatment on oxidative stress and inflammation after severe burn.[65] | Csontos | 2012 |
| 34. The effects of intravenous glutamine supplementation in severely burned, multiple traumatized patients.[66] | Cucerean-Badica | 2013 |
| 35. A comparison between occlusive and exposure dressing in the management of burn wound. | Dallal | 2016 |
| 36. Evaluation of the 'Early' Use of Albumin in Children with Extensive Burns: A Randomized Controlled Trial. | Dittrich | 2016 |
| 37. Interim pressure garment therapy (4–6 mmHg) and its effect on donor site healing in burn patients: study protocol for a randomised controlled trial.[67] | Donovan | 2016 |
| 38. Effect of whole body vibration on leg muscle strength after healed burns: a randomized controlled trial.[68] | Ebid | 2012 |
| 39. Effect of isokinetic training on muscle strength, size and gait after healed pediatric burn: a randomized controlled study.[69] | Ebid | 2014 |
| 40. Effect of 12 week isokinetic training on muscle strength in adult with healed thermal burn.[70] | Ebid | 2012 |
| 41. Effects of whole-body vibration exercise on bone mineral content and density in thermally injured children.[71] | Edionwe | 2016 |
| 42. Efficacy of platelet rich plasma application in comparison to conventional dressing therapy in partial thickness burn wound.[72] | Ehmer al Ibran | 2014 |
| 43. Effect of probiotic administration in the therapy of pediatric thermal burn.[73] | El-ghazely | 2016 |
| 44. Heparin/N-acetylcysteine: an adjuvant in the management of burn inhalation injury: a study of different doses.[74] | Elsharnouby | 2014 |
| 45. The effect of levamisole on mortality rate among patients with severe burn injuries.[75] | Fatemi | 2013 |
| 46. Impact of stress-induced diabetes on outcomes in severely burned children.[76] | Finnerty | 2014 |
| 47. Outcome of Burns Treated With Autologous Cultured Proliferating Epidermal Cells: A Prospective Randomized Multi-center Intra-patient Comparative Trial.[77] | Gardien | 2016 |
| 48. Randomized controlled trial of three burns dressings for partial thickness burns in children.[78] | Gee Kee | 2015 |
| 49. Topical petrolatum gel alone versus topical silver sulfadiazine with standard gauze dressings for the treatment of superficial partial thickness burns in adults: a randomized controlled trial.[79] | Genuino | 2014 |
| 50. HEPBURN - investigating the efficacy and safety of nebulized heparin versus placebo in burn patients with inhalation trauma: study protocol for a multi-center randomized controlled trial.[80] | Glas | 2014 |
| 51. A multi-center study on the regenerative effects of erythropoietin in burn and scalding injuries: study protocol for a randomized controlled trial.[82] | Gunter | 2013 |
| 52. Early rehabilitative exercise training in the recovery from pediatric burn.[83] | Hardee | 2014 |
| 53. Quality of pediatric second-degree burn wound scars following the application of basic fibroblast growth factor: results of a randomized, controlled pilot study.[84] | Hayashida | 2012 |
| 54. Long-term propranolol use in severely burned pediatric patients: a randomized controlled study.[85] | Herndon | 2012 |

Continued

**Table 1**  Continued

| Trial title | First author | Year of publication |
|---|---|---|
| 55. Reversal of growth arrest with the combined administration of oxandrolone and propranolol in severely burned children.[86] | Herndon | 2016 |
| 56. Cost-Effectiveness of Laser Doppler Imaging in Burn Care in The Netherlands: A Randomized Controlled Trial.[87] | Hop | 2016 |
| 57. Effect of music intervention on burn patients' pain and anxiety during dressing changes.[88] | Hsu | 2016 |
| 58. Low dose of glucocorticoid decreases the incidence of complications in severely burned patients by attenuating systemic inflammation.[89] | Huang | 2015 |
| 59. An assessment of early Child Life Therapy pain and anxiety management: A prospective randomised controlled trial.[90] | Hyland | 2015 |
| 60. Prospective, randomised controlled trial comparing Versajet™ hydrosurgery and conventional debridement of partial thickness paediatric burns.[91] | Hyland | 2015 |
| 61. Construction of skin graft seams in burn patients: A prospective randomized double-blinded study.[92] | Isaac | 2016 |
| 62. Multi-axis shoulder abduction splint in acute burn rehabilitation: a randomized controlled pilot trial.[93] | Jang | 2015 |
| 63. Glucose control in severely burned patients using metformin: An interim safety and efficacy analysis of a phase II randomized controlled trial.[94] | Jeschke | 2016 |
| 64. The effect of ketoconazole on post-burn inflammation, hypermetabolism and clinical outcomes.[95] | Jeschke | 2012 |
| 65. The Effect of Distraction Technique on the Pain of Dressing Change among 3–6 Year-old Children.[96] | Kaheni | 2016 |
| 66. Prospective randomize-controlled comparison between silicone plus herbal extract gel versus Aloe Vera gel for burn scar prophylaxis.[97] | Keorochana | 2015 |
| 67. Effects of Enteral Glutamine Supplementation on Reduction of Infection in Adult Patients with Severe Burns.[98] | Kibor | 2014 |
| 68. Effects of sustained release growth hormone treatment during the rehabilitation of adult severe burn survivors.[99] | Kim | 2016 |
| 69. Virtual reality for acute pain reduction in adolescents undergoing burn wound care: a prospective randomized controlled trial.[100] | Kipping | 2012 |
| 70. The effects of splinting on shoulder function in adult burns.[101] | Kolmus | 2012 |
| 71. Prospective study on burns treated with Integra, a cellulose sponge and split thickness skin graft: comparative clinical and histological study--randomized controlled trial.[102] | Lagus | 2013 |
| 72. Evaluation of an oxygen-diffusion dressing for accelerated healing of donor-site wounds.[103] | Lairet | 2014 |
| 73. Anti-inflammatory effect of taurine in burned patients.[104] | Lak | 2015 |
| 74. A randomized controlled pilot study comparing aqueous cream with a beeswax and herbal oil cream in the provision of relief from postburn pruritus.[105] | Lewis | 2012 |
| 75. Human acellular dermal matrix allograft: A randomized, controlled human trial for the long-term evaluation of patients with extensive burns.[106] | Li | 2015 |
| 76. Selective digestive decontamination attenuates organ dysfunction in critically ill burn patients.[107] | Lopez-Rodriguez | 2015 |
| 77. Results of a prospective randomized controlled trial of early ambulation for patients with lower extremity autografts.[108] | Lorello | 2014 |
| 78. Moist occlusive dressing (Aquacel(Â) Ag) versus moist open dressing (MEBO(Â)) in the management of partial-thickness facial burns: a comparative study in Ain Shams University.[109] | Mabrouk | 2012 |
| 79. Enhancement of burn wounds healing by platelet dressing.[110] | Maghsoudi | 2013 |
| 80. Effect of immune-enhancing diets on the outcomes of patients after major burns.[111] | Mahmoud | 2014 |

Continued

**Table 1** Continued

| Trial title | First author | Year of publication |
|---|---|---|
| 81. Silver-coated nylon dressing plus active DC microcurrent for healing of autogenous skin donor sites.[112] | Malin | 2013 |
| 82. The application of platelet-rich plasma in the treatment of deep dermal burns: A randomized, double-blind, intra-patient-controlled study.[113] | Marck | 2016 |
| 83. Clinical safety and efficacy of probiotic administration following burn injury.[114] | Mayes | 2015 |
| 84. Three donor site dressings in pediatric split-thickness skin grafts: study protocol for a randomised controlled trial.[115] | McBride | 2015 |
| 85. Evaluation of who oral rehydration solution (ORS) and salt tablets in resuscitating adult patients with burns covering more than 15% of total body surface area (TBSA).[116] | Moghazy | 2016 |
| 86. Efficacy and adverse events of early high-frequency oscillatory ventilation in adult burn patients with acute respiratory distress syndrome.[117] | Mohamed | 2016 |
| 87. Effect of amniotic membrane on graft take in extremity burns.[118] | Mohammadi | 2013 |
| 88. Comparison of the application of allogeneic fibroblast and autologous mesh grafting with conventional method in the treatment of third-degree burns.[119] | Moravvej | 2016 |
| 89. Effect of low-intensity laser on the neuropathic common peroneal nerve post burn.[120] | Mowafy | 2016 |
| 90. Clinical Efficacy Test of Polyester Containing Herbal Extract Dressings in Burn Wound Healing.[121] | Muangman | 2016 |
| 91. Effect of oral olive oil on healing of 10%–20% total body surface area burn wounds in hospitalized patients.[123] | Najmi | 2015 |
| 92. Double-blind, randomized, pilot study assessing the resolution of postburn pruritus.[124] | Nedelec | 2012 |
| 93. Comparing outcomes of sheet grafting with 1:1 mesh grafting in patients with thermal burns: a randomized trial.[125] | Nikkah | 2014 |
| 94. Comparison of hydrogel produced by radiation as applied at the research center (Yazd branch) with maxgel and routine dressing for second-degree burn repair in Yazd burn hospital.[126] | Noorbala | 2016 |
| 95. Effectiveness of cerium nitrate-silver sulfadiazine in the treatment of facial burns: a multi-center, randomized, controlled trial.[127] | Oen | 2012 |
| 96. Influences of purposeful activity versus rote exercise on improving pain and hand function in pediatric burn.[128] | Omar | 2012 |
| 97. Botulinum toxin and burn induces contracture.[130] | Omranifard | 2016 |
| 98. Results of a pilot multi-center genotype-based randomized placebo-controlled trial of propranolol to reduce pain after major thermal burn injury.[129] | Orrey | 2015 |
| 99. A proper enteral nutrition support improves sequential organ failure score and decreases length of stay in hospital in burned patients.[131] | Ostradrahimi | 2016 |
| 100. Topical silver sulfadiazine vs collagenase ointment for the treatment of partial thickness burns in children: a prospective randomized trial.[132] | Ostlie | 2012 |
| 101. Prospective randomized phase II Trial of accelerated re-epithelialization of superficial second-degree burn wounds using extracorporeal shock wave therapy.[133] | Ottomann | 2012 |
| 102. A randomized and controlled multi-center prospective study of the Chinese medicinal compound Fufang Xuelian Burn Ointment for the treatment of superficial and deep second-degree burn wounds.[134] | Ouyang | 2014 |
| 103. Prospective comparison of packed red blood cell-to-fresh frozen plasma transfusion ratio of 4: 1 vs 1: 1 during acute massive burn excision.[135] | Palmieri | 2012 |
| 104. A herbal cream consisting of Aloe Vera, Lavandulastoechas, and Pelargonium roseum as an alternative for silver sulfadiazine in burn management.[136] | Panahi | 2012 |
| 105. Interactive gaming consoles reduced pain during acute minor burn rehabilitation: A randomized, pilot trial.[137] | Parker | 2016 |
| 106. A Pilot Prospective Randomized Control Trial Comparing Exercises Using Videogame Therapy to Standard Physical Therapy: 6 Months Follow-Up.[138] | Parry | 2015 |

Continued

**Table 1** Continued

| Trial title | First author | Year of publication |
|---|---|---|
| 107. An open, prospective, randomized pilot investigation evaluating pain with the use of a soft silicone wound contact layer vs bridal veil and staples on split thickness skin grafts as a primary dressing.[139] | Patton | 2013 |
| 108. Effects of community-based exercise in children with severe burns: A randomized trial.[140] | Pena | 2015 |
| 109. Effects of propranolol and exercise training in children with severe burns.[141] | Porro | 2013 |
| 110. Five-year outcomes after oxandrolone administration in severely burned children: a randomized clinical trial of safety and efficacy.[142] | Porro | 2012 |
| 111. Clinical effectiveness, quality of life and cost-effectiveness of Flaminal versus Flamazine in the treatment of partial thickness burns: study protocol for a randomized controlled trial.[144] | Rashaan | 2016 |
| 112. Five-Year Outcomes after Long-Term Oxandrolone Administration in Severely Burned Children: A Randomized Clinical Trial.[145] | Reeves | 2016 |
| 113. A novel rapid and selective enzymatic debridement agent for burn wound management: a multi-center RCT.[146] | Rosenburg | 2013 |
| 114. Effects of cholecalciferol supplementation and optimized calcium intakes on vitamin D status, muscle strength and bone health: a 1 year pilot randomized controlled trial in adults with severe burns.[147] | Rousseau | 2015 |
| 115. Evaluation of Amniotic Membrane Effectiveness in Skin Graft Donor Site Dressing in Burn Patients.[148] | Salehi | 2015 |
| 116. A feasibility study assessing cortical plasticity in chronic neuropathic pain following burn injury.[143] | Santos Portilla | 2013 |
| 117. Perioperative treatment algorithm for bleeding burn patients reduces allogeneic blood product requirements.[149] | Schaden | 2012 |
| 118. A prospective clinical trial comparing Biobrane, Dressilk, and PolyMem dressings on partial-thickness skin graft donor sites.[150] | Schulz | 2016 |
| 119. Effectiveness of Aloe Vera gel compared with 1% silver sulphadiazine cream as burn wound dressing in second degree burns.[151] | Shahzad | 2013 |
| 120. The comparison between modified kligman formulation versus kligman formulation and intense pulsed light in the treatment of the post-burn hyperpigmentation.[152] | Siadat | 2016 |
| 121. A comparative study of spray keratinocytes and autologous meshed split-thickness skin graft in the treatment of acute burn injuries.[154] | Sood | 2015 |
| 122. Long-Term Administration of Oxandrolone Improves Lung Function in Pediatric Burned Patients.[155] | Sousse | 2016 |
| 123. An open, parallel, randomized, comparative, multicenter investigation evaluating the efficacy and tolerability of Mepilex Ag versus silver sulfadiazine in the treatment of deep partial-thickness burn injuries.[156] | Tang | 2015 |
| 124. Non-ablative fractional laser provides long-term improvement of mature burn scars - A randomized controlled trial with histological assessment.[157] | Taudorf | 2015 |
| 125. Fluid therapy lidco controlled trial - Optimization of volume resuscitation of extensively burned patients through noninvasive continuous real-time hemodynamic monitoring LiDCO.[158] | Tokarik | 2013 |
| 126. Burn donor site dressing using melolin and flexigrid versus conventional dressing.[159] | Vejdan | 2015 |
| 127. Laser Doppler imaging as a tool in the burn wound treatment protocol.[160] | Venclauskiene | 2014 |
| 128. Low-dose hydrocortisone reduces norepinephrine duration in severe burn patients: a randomized clinical trial.[161] | Venet | 2015 |
| 129. A Comparative Study of Paediatric Thermal Burns Treated with Topical Heparin and Without Heparin.[162] | Venkatachalapathy | 2014 |
| 130. Aquacel() Ag dressing versus ActicoatTM dressing in partial thickness burns: a prospective, randomized, controlled study in 100 patients. Part 1: burn wound healing.[24] | Verbelen | 2014 |
| 131. Skin stretching for primary closure of acute burn wounds.[163] | Verhaegen | 2014 |

Continued

**Table 1** Continued

| Trial title | First author | Year of publication |
|---|---|---|
| 132. Xbox KinectTM based rehabilitation as a feasible adjunct for minor upper limb burns rehabilitation: A pilot RCT.[164] | Voon | 2016 |
| 133. Local application of low-dose insulin in improving wound healing after deep burn surgery.[165] | Wang | 2016 |
| 134. Gabapentin is ineffective as an analgesic adjunct in the immediate postburn period.[166] | Wibbenmeyer | 2014 |
| 135. A prospective randomised clinical pilot study to compare the effectiveness of Biobrane (R) synthetic wound dressing, with or without autologous cell suspension, to the local standard treatment regimen in paediatric scald injuries.[167] | Wood | 2012 |
| 136. Effective symptomatic treatment for severe and intractable pruritus associated with severe burn-induced hypertrophic scars: A prospective, multicenter, controlled trial.[168] | Wu | 2016 |
| 137. Propranolol reduces cardiac index but does not adversely affect peripheral perfusion in severely burned children.[169] | Wurzer | 2016 |
| 138. A new method of microskin autografting with a Vaseline-based moisture dressing on granulation tissue.[170] | Xiao | 2014 |
| 139. Recombinant human granulocyte-macrophage colony-stimulating factor hydrogel promotes healing of deep partial thickness burn wounds.[171] | Yan | 2012 |
| 140. A comparative study of the dressings silver sulfadiazine and Aquacel Ag in the management of superficial partial-thickness burns.[172] | Yarboro | 2013 |
| 141. A clinical trial designed to evaluate the safety and effectiveness of a thermosensitive hydrogel-type cultured epidermal allograft for deep second-degree burns.[173] | Yim | 2014 |
| 142. Study of the use of recombinant human granulocyte-macrophage colony-stimulating factor hydrogel externally to treat residual wounds of extensive deep partial-thickness burn.[174] | Yuan | 2015 |
| 143. Effect of Olea ointment and Acetate Mafenide on burn wounds - A randomized clinical trial.[175] | Zahmatkesh | 2015 |
| 144. Effects of puerarin on the inflammatory role of burn-related procedural pain mediated by $P2 \times 7$ receptors.[176] | Zhang | 2013 |
| 145. Effects of early enteral nutrition on the gastrointestinal motility and intestinal mucosal barrier of patients with burn-induced invasive fungal infection.[81] | Zhang | 2016 |
| 146. Maximizing the safety of glycerol preserved human amniotic membrane as a biological dressing.[177] | Zidan | 2015 |
| 147. Therapeutic Value of Blood Purification and Prognostic Utilities of Early Serum Procalcitonin, C Reactive Protein, and Brain Natriuretic Peptide Levels in Severely Burned Patients with Sepsis.[178] | Zu | 2015 |

A total of 9022 participants were recruited across the 140 studies (study protocols not included n=7). The number of patients recruited per trial ranged from 3 to 612 (median 50; IQR 30–88) for full RCTs and from 10 to 52 (median 21; IQR 16–28) for pilot studies. 50.4% of full RCTs recruited fewer than 50 participants (table 2). The majority (89.7%) of studies recruited (or planned to recruit) participants on one site alone. Of the 10.2% (15) of studies that were multicentre, nine (60%) undertook research at only two or three sites. Thirty-two countries from the six global regions recruited patients into the 147 RCTs (table 2). The country that undertook the most studies was the USA with 22.4% (33), followed by Iran with 12.9% (19) and China with 10.9% (16) of published studies. Of the 32 countries, 59.3% (19) published only one trial in this time period. The most common trial interventions related to dressings and wound care 29.2%

(43), followed by surgical technique 11.6% (17) and management of pain and itch 10.9% (16) (table 2).

## Outcomes
A total of 1494 clinical outcomes were reported of which, after de-duplication, 955 different, unique outcomes remained. Of the 1494 outcomes reported, 27.7% (421) were common across two studies or more. Of *these* outcomes, 50.3% (78) appear in only two trials and 84.5% appear in five trials or fewer. The number of outcomes reported per trial varied from one to 37 (median 9; IQR 5,13) (table 3). No single outcome was reported across all 147 studies.

*Outcome definition variation:* Outcomes assessing the same healthcare issue were commonly defined differently. An example is burn wound healing which was defined in 166 different ways. Examples include: healing percentage

| Table 2 Randomised controlled tiral (RCT) detail | |
|---|---|
| | **Studies** |
| Number of RCTs | 127/147 (86.4) |
| Number of pilot studies | 13/147 (8.8) |
| Number of RCT protocols | 7/147 (4.8) |
| World region for recruitment | |
| Asia | 54/147 (36.7) |
| North America | 36/147 (24.5) |
| Europe | 26/147 (17.7) |
| Africa | 15/147 (8.5) |
| Latin America | 1/147 (0.7) |
| Australasia | 15/147 (8.5) |
| Year published | |
| 2012 | 26/147 (17.8) |
| 2013 | 24/147 (16.3) |
| 2014 | 24/147 (16.3) |
| 2015 | 33/147 (22.4) |
| 2016 | 40/147 (27.2) |
| Number of sites | |
| 1 | 132/147 (89.8) |
| 2–3 | 9/147 (6.1) |
| 4–5 | 2/147 (1.4) |
| >5 | 4/147 (2.7) |
| Number of participants in full RCTs | |
| <10 | 4/127 (3.1) |
| 11–50 | 62/127 (48.8) |
| 51–100 | 39/127 (30.7) |
| 101–150 | 11/127 (8.7) |
| >150 | 11/127 (8.7) |
| Participants recruited | |
| <18 years | 24/147 (16.3) |
| >18 years | 59/147 (40.1) |
| Mixed age range | 25/147 (17.0) |
| Not stated | 34/147 (23.1) |
| N/A (protocol)‡ | 5/147 (3.4) |
| Type of intervention | |
| Dressings and wound care | 38/147 (25.9) |
| Surgical technique | 19/147 (12.9) |
| Treatment of pain or itch* | 16/147 (10.9) |
| Impact of exercise and rehabilitation | 13/147 (8.8) |
| Intensive care management | 10/147 (6.8) |
| Treatment of hypermetabolism | 8/147 (5.4) |
| Nutrition | 8/147 (5.4) |
| Scar management | 7/147 (4.8) |
| Treatment of inhalational injury | 3/147 (2.0) |
| Use of topical rHGM | 3/147 (2.0) |
| Use of rHGH | 3/147 (2.0) |

Continued

| Table 2 Continued | |
|---|---|
| | **Studies** |
| Sugar management | 2/148 (2.0) |
| Treatment of infection | 2/147 (1.4) |
| Treatment of DVT | 2/147 (1.4) |
| Blood management | 2/147 (1.4) |
| Extracorporeal shock wave therapy | 2/147 (1.4) |
| Platelet-rich plasma use | 2/147 (1.4) |
| Others† | 7/147 (5.4) |

*Inc. distraction for dressing changes.
†Inc. levamisole, hyperbaric oxygen, fibroblast growth factor, oral calcium use, ketoconazole, low intensity laser.
‡No participants reported as study is a protocol.
DVT, deep vein thrombosis; N/A, not applicable; rHGH, recombinant human growth hormone; rHGM, recombinant Human Granulocyte-Macrophage colony-stimulating factor.

at specified timepoints, incidence of complete wound healing, incidence of 30% wound healing and length of time until 50% epithelialisation of burn wound. Similar variation in definition of burn wound infection existed with 79 unique outcome definitions including: bacterial colonisation of burn wound, days of antibiotics, incidence of local infection, incidence of positive wound cultures, periwound redness, rate of bacterial clearance from wound and number of inflammatory cells in the wound.

*Outcome timing variation*: There were 2743 outcomes measured if the same outcome measured at different timepoints across all the 147 RCTs are included; for example, size of burn wound measured at 1 week and again at 2 weeks, were recorded as different outcomes for this exercise. Of these, 76.9% (2109) were assessed at less than 6 months after injury, 16.6% (456) were measured after 6 months and before 3 years after injury, and only 5.1% (140) were measured at more than 3 years after injury (figure 2). The timing of outcome measurement was not reported for 38 outcomes.

*Outcome domains*: The 955 *different* clinical outcomes were organised into 54 domains (groups of similar outcomes). Table 4 categorises the domains into overarching categories and gives examples and total numbers of outcomes within each domain.

| Table 3 Reported outcomes | |
|---|---|
| **Number of outcomes per study** | |
| 1 | 4/147 (27.2%) |
| 2–5 | 34/147 (23.1%) |
| 6–10 | 53/147 (36.1%) |
| 11–20 | 41/147 (27.9%) |
| >20 | 15/147 (10.2%) |

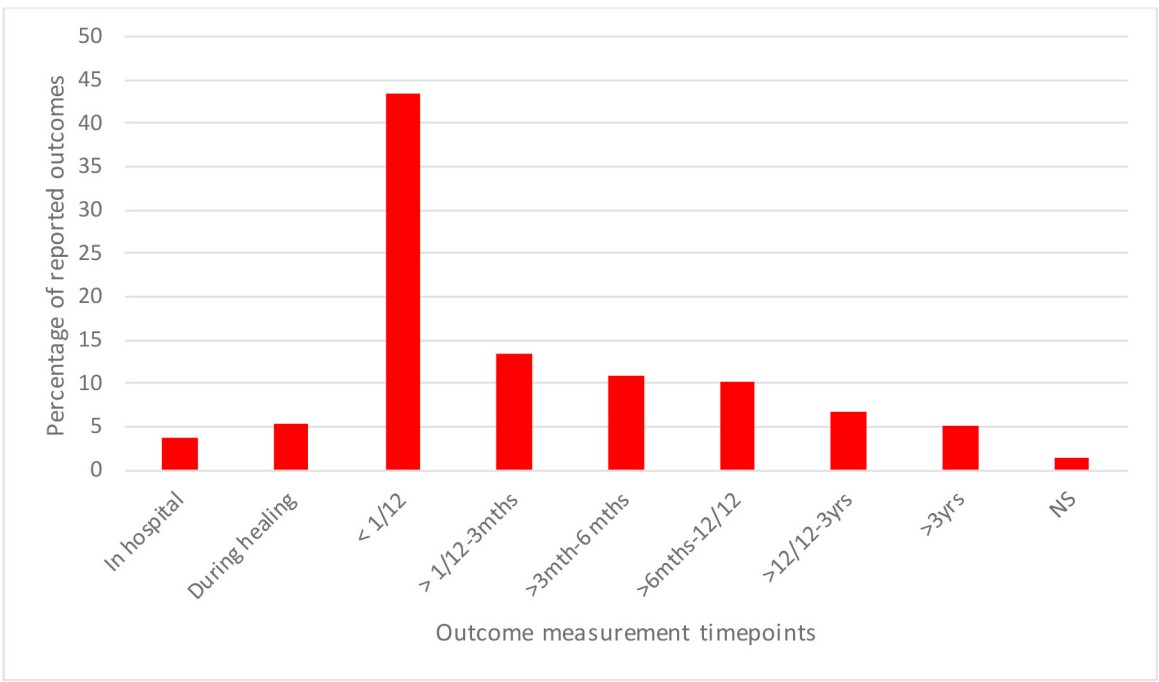

**Figure 2** Reported timepoint at which outcome was assessed. mths, months; NS, not stated; yrs, years.

## DISCUSSION

This systematic review aimed to examine outcome reporting in RCTs in burn care. Of the 147 included studies, 1494 outcomes were identified with 955 of these unique. There was overlap in terminology, inconsistent definitions and variation in time after injury at which the outcomes were measured. Only 30% of the outcomes reported were included in more than one study. There was no single outcome reported across all 147 trials. Commonly-reported outcomes were defined differently between trials, such as burn wound healing which was defined in 166 different ways. Such heterogeneity of outcome reporting across trials will limit evidence synthesis and result in research wastage.

The findings in this review have been seen elsewhere in the burns-specific and other clinical literature. A Cochrane review of 30 RCTs concluded that it was impossible to draw conclusions about burn dressing effectiveness, as the trials evaluated a variety of clinical outcomes.[18 179] Over the same period as this review, nine Cochrane reviews have had direct relevance to the management of patients with burns.[18 180–187] None could draw firm conclusions due to methodological issues including heterogeneity of outcome reporting. Heterogeneity is found in the reporting of outcomes relating to critical care, neurological illness, breast reconstruction surgery, prostate cancer, hip and knee replacement, oesophagectomy surgery, low back pain and in cardiac arrest trials among others.[188–195] Variation in the definitions of outcomes has also been found within published studies of other healthcare areas. A systematic review of 90 papers reporting wound infection after general surgical procedures identified 41 definitions for wound infection

itself, including three published by expert groups.[196] Similarly, a total of 56 definitions were identified from 97 studies reporting anastomotic leak rates after gastrointestinal surgery despite publication of a standard definition 2 years before the beginning of the review.[197]

In this review, we identified and agreed the grouping of the 955 unique outcomes into 54 outcome domains. There is no agreement between COS reviewers about how best to classify outcomes into domains. Williamson published a taxonomy of categorising outcome domains.[198] Other authors have suggested different ways of doing this, all addressing different needs[32 33 199]). In the Williamson taxonomy, the authors state that of 99 COS studies, 21 applied their own approach to outcome classification and only six used an existing system. As we had identified a large number of different clinical burn outcomes and as the outcomes we extracted did not clearly fall within the Williamson taxonomy, we decided to use our own approach to domain classification. We used five multidisciplinary researchers and a patient working independently, and subsequently together, to bring different views and as little bias as possible to the process.

A solution to the above described variation in outcome reporting across trials, is the development of a COS.[21 200 201] A COS is a minimum set of the most important outcomes, agreed and recommended for measurement in all trials for a particular condition.[31 32] While not limiting choice, a COS will prespecify a set of outcomes to ensure consistency of reporting and the ability to collate evidence into systematic reviews by allowing researchers to compare 'like with like'.[33] Trials can still select additional outcomes in addition to the minimum core set. This approach has been shown to improve the consistency of outcome reporting.[202 203] Although there

**Table 4** Outcome category, domains and examples of outcomes

| Outcome category | Outcome domain | Outcome examples | No of unique outcomes per domain |
|---|---|---|---|
| Patient-reported | Ability to carry out daily tasks | Functional level of independence | 1 |
| | Anxiety about medical procedures and appointments | Pain anxiety<br>Anxiety before dressing changes | 4 |
| | Generalised anxiety | General anxiety | 1 |
| | Appearance | Facial symmetry<br>Overall scar appearance | 3 |
| | Blister fluid | Amount of exudate | 3 |
| | Burden of care | Frequency of dressing changes<br>Time taken for daily wound cleaning | 7 |
| | Comfort of dressings | Dressing comfort | 1 |
| | Psychological well-being | Improvement in well-being | 1 |
| | Mental ability | | 2 |
| | Quality and quantity of sleep | Quantity of sleep<br>Incidence of sleep disturbance | 17 |
| | Effect of scar on movement (contractures) | Cognitive performance | 3 |
| | Return to work/school or previous function | Return to work or previous function | 1 |
| | Burn wound pain | Wound pain intensity at baseline<br>Pain tolerance | 29 |
| | Donor site pain | Donor site pain at rest<br>Donor site pain while walking | 6 |
| | Pain during procedures | Wound pain at dressing changes<br>Pain during hydrotherapy | 14 |
| | Scar pain | Functional scar pain<br>Incidence of neuropathic pain | 13 |
| | Itch | Baseline pruritus<br>Itch severity reduction | 24 |
| Pathophysiological | Ability to fight infection | Change in IgA<br>IL-1beta in blood<br>Serum interferon gamma levels | 36 |
| | Body weight maintenance | Incidence of weight loss<br>Body weight decrease from baseline | 26 |
| | Bone strength | Bone mineral density<br>Incidence of osteoporosis | 30 |
| | Breathing and lungs | Forced expiratory volume in 1 s<br>Functional residual capacity | 27 |
| | Donor site problems after healing | Donor site pigmentation<br>Sensation of donor site | 24 |
| | Effect of burn on genes | Gene expression patterns | 1 |
| | Effect of burn on how the body uses energy | Change in percentage of predicted resting energy expenditure | 2 |
| | Effect on heart and blood circulation | Incidence of cardiomegaly<br>Number of patients requiring norepinephrine | 28 |
| | Fitness | Maximum aerobic capacity<br>Exercise maximum minute ventilation | 12 |
| | Growth in children | Duration of growth arrest<br>Percentage change in height | 10 |
| | How well muscles work | Facial mimic function<br>Change in muscle function | 9 |
| | Mobility | Stride length<br>Knee range of motion | 22 |
| | Kidney function | Incidence of acute kidney injury<br>Requirement for renal replacement therapy | 17 |
| | Liver function | Hepatic function<br>Ammonia levels | 11 |

Continued

**Table 4** Continued

| Outcome category | Outcome domain | Outcome examples | No of unique outcomes per domain |
|---|---|---|---|
| | Medical tests to indicate how unwell a patient is | Albumin level<br>Change in pH | 84 |
| | More than one organ failing (multiorgan failure) | Incidence of multi organ failure<br>Percentage of patients with organ dysfunction | 7 |
| | Muscle strength | Knee extensor strength<br>Hamstring strength adjusted for body weight | 30 |
| | Stomach and bowel function | Days of diarrhoea<br>Incidence of abdominal distension | 13 |
| | Burn wound healing | Burn wound area at timepoints<br>Days until wound closure | 166 |
| | Donor site healing | Donor site healing to 90%<br>Time to donor site re-epithelialisation | 9 |
| Complications | Complications of drug treatment | Adverse drug reactions<br>Allergic dermatitis | 52 |
| | Blood product transfusion | Blood transfused per kg during hospitalisation<br>Total volume FFP transfused | 11 |
| | Burn wound infection | Wound bacterial colonisation<br>Wound contamination postoperatively | 80 |
| | Death from burn injury | Mortality related to burn size | 1 |
| | Death from any cause | Overall mortality<br>In-hospital mortality | 14 |
| | Effects of fluid from a drip | Incidence of fluid creep<br>Net fluid balance at specified times | 17 |
| | Infections other than burn wound infection | Incidence of central catheter related infections<br>Pulmonary infection | 7 |
| | Sepsis | Days of sepsis<br>Incidence of positive blood cultures | 7 |
| Scar-related | Scar colour | Erythema index<br>Scar melanin levels | 25 |
| | Scar texture | Scar height<br>Change in scar distensibility | 47 |
| | Scar size | Scar surface area | 1 |
| | Treatment for scars | Numbers of patients assessed for scar management<br>Numbers of patients needing scar management | 2 |
| Healthcare-related | Costs of treatment for NHS/hospital | Costs of analgesics for dressing changes<br>Pressure garment costs | 14 |
| | Length of hospital stay | Length of stay adjusted for burn size<br>Days in hospital | 7 |
| | Length of stay in intensive care unit | Length of ICU stay | 3 |
| | Length of time on life support machine | Duration of mechanical ventilation | 1 |
| | Use of medicines to treat symptoms | Pain relief required during dressing changes<br>Opioid consumption | 12 |

FFP, fresh frozen plasma; ICU, intensive care unit; IL, interleukin; NHS, National Health Service.

is no COS for burn care, work was undertaken in 2008 to agree a set of burn outcome domains.[198] However, the work was undertaken by a small group of clinicians, lacked patient involvement and reported little methodological detail.[204] Considerable work to develop COS methodology has also been undertaken since this publication.[205 206] The COMET Initiative disseminates resources for COS development and supports methodological developments in this area.[207 208] COMET recommends a four-step process to develop a COS: (a) agreement of the scope, (b) assessment of the need, (c) development of a protocol and finally (d) agreement of the COS.[15] This systematic review has satisfied the first two phases for the development of a burn care COS. The final phase encompasses organising a comprehensive long-list of all potential outcomes into domains (of which the clinical domains for burn care are listed in table 4) and prioritising these domains using a consensus process.[209–211]

The strengths of this review are that the protocol and data extraction proforma were prespecified and the literature search was systematic and comprehensive, including

four major healthcare trial databases. To account for multidisciplinary perspectives, two researchers, two clinicians and a patient were involved in the domain process. It is also novel because it is the first to demonstrate, in detail and using systematic methodology, the scale of the heterogeneity of outcome reporting in burn care research. Limitations include the exclusion of publications in languages other than English. However, international publications were included to reduce the risk of selection bias. The search was also time-limited which may have excluded outcomes from older studies. The reason for the time limitation was to identify research relevant to modern burn care. The search was also limited to trials reporting clinical outcomes. Other work is in progress to assess patient-reported outcomes in burn care research. This was a review undertaken systematically to a prespecified protocol. However, a formal quality assessment of studies was not undertaken, as we were researching the reporting of outcomes and not attempting to analyse the effects of interventions. A COS for burn care research would address the issue of heterogeneity of outcome reporting between trials, lead to research that is more likely to measure relevant outcomes, enhance the value of burn care systematic reviews and reduce research waste.

## CONCLUSION

We have shown that multiple different unique outcomes are reported in trials of burn care interventions. Different definitions are used to assess the same outcome issue and outcomes are measured at different time points after injury. This heterogeneity and inconsistency in outcome reporting prevent effective evidence synthesis and limits the quality of evidence available for clinical decision-making. Our review demonstrates that until greater consistency is achieved in outcome reporting in trials, it is unlikely that clinicians will be able to synthesise evidence across studies to understand the effects of surgical and non-surgical treatments following burn injury. It is recommended that a burn care COS is developed to support the effective synthesis of trial data and allow more informed clinical decision-making for the benefit of patients.

**Acknowledgements** We would like to thank Dr Jason Wasiak, Senior Research Fellow, University of Melbourne, Monash University, Medical School, for his invaluable help in critically editing the paper. We would like to also thank Ms Joanna Hooper, senior outreach librarian and the University Hospitals Bristol NHS Foundation Trust medical library for their help in accessing articles.

**Contributors** AEY wrote the paper and conceived the project with the support of JMB. JMB and SBr edited and critically revised the article. AD contributed to data extraction and edited the paper. SBI assisted in the domain name choice and article editing and readability. All authors have read and approved the manuscript.

**Funding** This article/paper/report presents independent research funded by the National Institute for Health Research (NIHR) Doctoral Research Fellowship DRF-2016-09-031. JMB is part-funded by the Medical Research Council ConDuCT-II Hub (Collaboration and innovation for Difficult and Complex randomised controlled Trials In Invasive procedures—MR/K025643/1). The study was also supported by the NIHR Biomedical Research Centre at the University Hospitals Bristol NHS Foundation Trust and the University of Bristol.

**Disclaimer** The views expressed are those of the author(s) and not necessarily those of the NHS, the NIHR or the Department of Health.

**Competing interests** JB is an NIHR Senior Investigator. All other authors declare no competing interests.

**Patient consent for publication** Not required.

**Ethics approval** South West Frenchay Research Ethics Committee (ref: 17/SW/0025 IRAS 221625).

**Provenance and peer review** Not commissioned; externally peer reviewed.

**Data sharing statement** No further data are available.

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
