## [Reviewer comments · BMJ Open]

ARTICLE DETAILS

TITLE (PROVISIONAL)	A systematic review of clinical outcome reporting in randomised controlled trials of burn care.
AUTHORS	Young, Amber; Davies, Anna; Bland, Sophie; Brookes, Sara; Blazeby, Jane

VERSION 1 – REVIEW

REVIEWER	Ewelina Rogozinska Queen Mary University of London, UK
REVIEW RETURNED	16-Oct-2018

GENERAL COMMENTS	The authors Young et al. provided an overview of clinical outcome reporting in trials on burn care interventions. The study applied a systematic review methodology to identify and assess the type of clinical outcomes used in the trials published between 2012 and 2016. Only after reading the following abstract questions come to my mind - why authors decided to narrow their search to only four years? I see later on in the manuscript that this period is supposed to reflect "the modern burn care", however, I find this a quite strong limitation. Some of the sentences in the abstract seem quite randomly chucked in without sufficient level of detail, e.g. "Variation in outcome reporting was assessed across trials." (how?) "Outcomes were classified into domains." (using what classification? what for?) Without a context it is difficult to make sense of the information in result part: "Numbers of unique outcomes per trial varied from one to 37 (median 9; IQR 8)." How big were those trials? "No outcome was reported in all studies." Unclear sentence - none of the 995 extracted outcomes appeared in all 147 trials? Also, is this a realistic expectation & need to have 100% coverage? "Outcomes were classified into 54 domains" as we don't know much about the classification system it is hard to interpret this number and its significance. In the conclusions, we read "This review has demonstrated heterogeneity in selection, definitions and timing of outcome reporting in burn care trials." While there is very little info on the timing of measurements and nothing about the heterogeneity in definitions." In the main manuscript again we read "Of the 147 included studies, 1,494 outcomes were identified, with overlap in terminology, inconsistent definitions and variation in time after injury at which the outcomes were measured." While there is no
---

	information on the outcomes definitions/terminology. What is the significance of Cohen's Kappa indicator for inter-rater reliability of study selection? Haven't the authors ultimately discussed the discrepancies and sought a third reviewer opinion regarding study's eligibility? Minor remarks The manuscript is written mostly in passive and very impersonal format. Introduction of active voice and better flow between the sentences would benefit its readability. Please check the manuscript for the use of abbreviation - I noticed "radomised controlled trial" being used despite early introduction of "RCT" abbreviation in the manuscript. Excel is spelt with a capital letter. Overall, I feel the paper has a potential but requires improvements as in the current form does not make a strong case for COS. I very much applaud the authors on the active involvement of the patient and public representatives in their work.
--	---

REVIEWER	Clifford Sheckter Stanford University, USA
REVIEW RETURNED	17-Oct-2018

GENERAL COMMENTS	1) This study tackled a large topic. The burn community would benefit for more standardized trial outcomes reporting, especially with regard to patient reported outcomes. 2) The outcomes domain distillation process should be grounded. Did you use a Delphi adjacent process? Were the reviewers blinded to each other? 3) Outcomes. I'm struggling a bit with the domains section. I know you wanted to condense all these studies into nicely packaged domains, but you lose too much granularity in the process. For mortality, it's simple, because there is no interpretation needed. For more subjective outcomes such as "mental ability," "psychological wellbeing," "complications of treatment," the reader has little understanding of what the trials were investigating. I'd like to know what outcomes they actually used. I would appreciate a thorough redesign of this section. For example, it would be nice to have headings such as "Patient Reported Outcomes," "Physiologic Outcomes," "Scar," etc. You can then list sub-domains underneath theses. You can included examples within each (e.g. with scar outcomes Vancouver scar score, etc.). These are just suggestions--do as you wish. However, this table is pivotal for the manuscript, and as of now it's lacking in organization and granularity.
--

VERSION 1 – AUTHOR RESPONSE

Reviewer: 1

Reviewer Name: Ewelina Rogozinska

Institution and Country: Queen Mary University of London, UK

Please state any competing interests or state 'None declared': None declared

The authors Young et al. provided an overview of clinical outcome reporting in trials on burn care interventions. The study applied a systematic review methodology to identify and assess the type of clinical outcomes used in the trials published between 2012 and 2016. Only after reading the following abstract questions come to my mind - why authors decided to narrow their search to only four years? I see later on in the manuscript that this period is supposed to reflect "the modern burn care", however, I find this a quite strong limitation.

Thank you and we understand your concern regarding the time limitation to the review. As described above: this systematic review was undertaken to extract outcomes reported in burn care trials and not to assess the effect of an intervention. A recent five-year time period was chosen to allow enough studies to be selected in a systematic way that would demonstrate whether heterogeneity of outcome reporting was occurring and whether there was a need for a Core Outcome Set (COS) in burn care research. It was necessary for us to stop at the end of December 2016 to use the results of the review to inform a Delphi study in line with the funders (NIHR) timescale. We believe that if we were to expand the systematic review for another six months or year, we would not demonstrate any more clearly the heterogeneity in outcome reporting and the need for a COS in burn care research. We hope this is acceptable to the BMJ Open editorial team and the reviewers.

Some of the sentences in the abstract seem quite randomly chucked in without sufficient level of detail, e.g. "Variation in outcome reporting was assessed across trials." (how?) "Outcomes were classified into domains." (using what classification? what for?) We apologise if this was not clear in the abstract. We have added more details to clarify the sentences detailed above.

Without a context it is difficult to make sense of the information in result part: "Numbers of unique outcomes per trial varied from one to 37 (median 9; IQR 8)." How big were those trials? The trials varied in size from one to more than five recruiting sites (Table 2), with numbers of participants ranging from < 10 to more than 150. There was no relationship between trial size - based on number of sites and participants - and numbers of reported outcomes.

"No outcome was reported in all studies." Unclear sentence - none of the 995 extracted outcomes appeared in all 147 trials? Also, is this a realistic expectation & need to have 100% coverage? We have re-worded this statement and agree that it was previously unclear.

"Outcomes were classified into 54 domains" as we don't know much about the classification system it is hard to interpret this number and its significance. We agree it is difficult to provide a context for this in the abstract. Some more detail has been added to the methods and results on the abstract to make this clearer. We have added the numbers of unique outcomes per domain in order to give some context. We have also re-worded this section in the methodology section of the main body of the research.

In the conclusions, we read "This review has demonstrated heterogeneity in selection, definitions and timing of outcome reporting in burn care trials." While there is very little info on the timing of measurements and nothing about the heterogeneity in definitions." It has been made clearer by re-labelling the relevant sections and adding more detail in the results regarding timing of outcome reporting and variation in outcome definitions.

In the main manuscript again we read "Of the 147 included studies, 1,494 outcomes were identified, with overlap in terminology, inconsistent definitions and variation in time after injury at which the outcomes were measured." While there is no information on the outcomes definitions/terminology.

Thank you. Please see above answer.

What is the significance of Cohen's Kappa indicator for inter-rater reliability of study selection? Haven't the authors ultimately discussed the discrepancies and sought a third reviewer opinion regarding study's eligibility?

This is a good point. Thank you. We have removed this statement and the reviewer is correct about the methodology we used.

Minor remarks

The manuscript is written mostly in passive and very impersonal format. Introduction of active voice and better flow between the sentences would benefit its readability.

Thank you. We have added some text with the active voice.

Please check the manuscript for the use of abbreviation - I noticed "radomised controlled trial" being used despite early introduction of "RCT" abbreviation in the manuscript.

Excel is spelt with a capital letter.

Altered with apologies – the reviewer is correct.

Overall, I feel the paper has a potential but requires improvements as in the current form does not make a strong case for COS. I very much applaud the authors on the active involvement of the patient and public representatives in their work.

Thank you.

Reviewer: 2

Reviewer Name: Clifford Sheckter

Institution and Country: Stanford University, USA

Please state any competing interests or state 'None declared': none

Please leave your comments for the authors below

1) This study tackled a large topic. The burn community would benefit for more standardized trial outcomes reporting, especially with regard to patient reported outcomes. We entirely agree. Thank you.

2) The outcomes domain distillation process should be grounded. Did you use a Delphi adjacent process? Were the reviewers blinded to each other?

The distillation of the outcomes into domains was a challenge of this and other similar studies. We were not able to use a published classification system, as none fitted the wide variety of burn care outcomes we extracted from the systematic review. We introduced as much objectivity and as little bias as possible in the process by involving reviewers of different clinical and research backgrounds including a patient. The distillation was undertaken independently (the reviewers were blinded to others' responses in the first stage) and subsequently undertaken together. It was an iterative process over a number of months. We agree that this needs to be justified in greater detail. We have added some text to explain this approach within the body of the paper with supporting references.

We apologise hugely, but do not know the term 'delphi-adjacent process'. We are sorry about this and are happy to respond if the reviewer would very kindly explain this term for us. Thank you.

3) Outcomes. I'm struggling a bit with the domains section. I know you wanted to condense all these studies into nicely packaged domains, but you lose too much granularity in the process. For mortality, it's simple, because there is no interpretation needed. For more subjective outcomes such as "mental ability," "psychological wellbeing," "complications of treatment," the reader has little understanding of what the trials were investigating. I'd like to know what outcomes they actually used. I would appreciate a thorough redesign of this section. For example, it would be nice to have headings such

as "Patient Reported Outcomes," "Physiologic Outcomes," "Scar," etc. You can then list sub-domains underneath these. You can included examples within each (e.g. with scar outcomes Vancouver scar score, etc.). These are just suggestions--do as you wish. However, this table is pivotal for the manuscript, and as of now it's lacking in organization and granularity.

Thank you for pointing this out. We agree with the reviewer. We have re-done this section and added the Table as you have suggested. Thank you.

VERSION 2 – REVIEW

REVIEWER	Clifford Sheckter Stanford University, USA
REVIEW RETURNED	25-Nov-2018
GENERAL COMMENTS	Thank you for reorganizing and adding granularity to the main table. This greatly improves the manuscript. You addressed the remainder of my comments.